# Extended 2D Consensus Hippocampus Segmentation

**Diedre Carmo**                                                    DIEDRE@DCA.FEE.UNICAMP.BR
**Leticia Rittner**                                                 LOTUFO@DCA.FEE.UNICAMP.BR
**Roberto Lotufo**                                                 LRITTNER@DCA.FEE.UNICAMP.BR
*MICLab, School of Electric and Computer Engineering, University of Campinas, Brazil*

**Bruna Silva**                                                    BRUNAF.10@HOTMAIL.COM
**Clarissa Yasuda**                                                CYASUDA@UNICAMP.BR
*Faculty of Medical Sciences, University of Campinas, Brazil*

## Abstract

Hippocampus segmentation plays a key role in diagnosing various brain disorders. Nowadays, segmentation is a manual, time consuming task and considered to be the gold-standard when evaluating automated methods. For years the best performing automatic methods were multi atlas based with 80 to 85% DICE and time consuming, but machine learning methods are recently rising with promising time and accuracy performance. In this work, a novel method for hippocampus segmentation is presented, based on the consensus of tri-planar U-Net inspired CNNs, with some modifications based on successful CNNs of the literature, and a patch extraction technique employing data from neighbor patches. Our in-house dataset has hippocampus atrophies resulted from epilepsy surgery treatment. Our method (labeled e2dhipseg) achieves cutting edge performance of 96% DICE in our test data. Our method was also compared to other recent methods in the public ADNI and HARP datasets.

**Keywords:** Deep Learning, Hippocampus Segmentation

## 1. Introduction and Related Work

The hippocampus can be atrophied when affected by neurodegenerative diseases (Andersen, 2007). In some cases surgical intervention to the hippocampus is necessary (Yasuda et al., 2010), and hippocampus segmentation is often used in the planning phase. Manual segmentation is the gold standard, even though inter rater variability is a concerning problem (Souza et al., 2018).

For a long time, the state-of-the-art in automated hippocampus segmentation was composed by multi-atlas methods (Wang et al., 2013; Iglesias and Sabuncu, 2015; Pipitone et al., 2014; Fischl, 2012), with around 0.8 to 0.9 DICE (Duarte et al., 1999). With the rise of Deep Learning, many studies attempt to employ it for hippocampus segmentation (Wachinger et al., 2018; Thyreau et al., 2018; Xie and Gillies, 2018; Chen et al., 2017), with similar or better DICE and, more importantly, reducing processing time from hours to seconds. Another related work is (Lucena et al., 2018), a skull stripping method that inspired our consensus strategy that involves the use of multiple CNNs performing segmentation over different MRI orientations, merged into a single final volume.

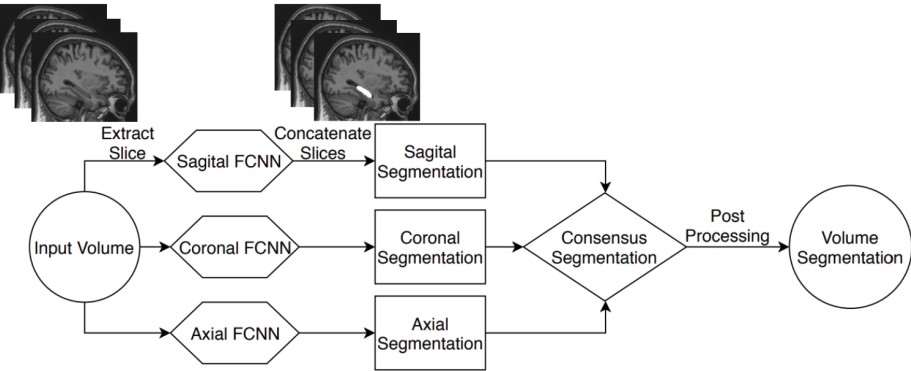

Figure 1: An outline of our method. Our fully convolutional neural networks (FCNNs) analyse the volume slice by slice, and post processing is done on the consensus volume.

## 2. Data and Methodology

The main private dataset used on this work was collected inhouse. Our dataset contains 214 MNI152 registered T1 weighted 3T MRI acquisitions. 142 are from patients of surgical treatment of epilepsy, where the hippocampus is not in its expected shape, with some atrophies. Hold-out was employed with 80% for training, 10% for validation and 10% for testing. Pre processing consists of minmax normalization of values. This research is approved by the UNICAMP Ethics and Research Committee (under CEP 1191/2011). For more extensive validation, we used 50 random ADNI(Petersen et al., 2010) isometric volumes, with balanced CN, MCI and AD cases. We also used the full HARP (Boccardi et al., 2015) 135 volumes. It is important to note that ADNI and HARP were not registered to MNI152 space.

Our methodology, consists of evaluating the consensus of volumes generated by three separate 3-channel 2D U-Net (Ronneberger et al., 2015) FCNNs, with encoders initialized with VGG11 weights (Simonyan and Zisserman, 2014) and residual connections blocks (He et al., 2016). The networks are trained on 3x64x64 patches (a central patch and its neighbours) on each brain orientation; sagital, coronal and axial, and their results are merged in a final consensus volume. Hyperparameters were selected after a grid-search, using SGD as an optimizer and DICE Loss. More details can be found in our pre-print (Carmo et al., 2019). The 3-channel 2D slices approach are labeled Extended 2D (Pereira et al., 2019). Simultaneous, independent work in (Ataloglou et al., 2019) used a similar initial strategy, although not using a U-Net like architecture and different post and pre processing steps.

In the test phase 160x160 center crop slices are segmented in their respective orientations (Figure 1), and concatenated in a volumetric mask per orientation. Pre registration is not needed. To generate a final consensus heatmap, each volume is given equal weight of 1/3 and the activation are summed. Binarization of the consensus volume is performed with a threshold of 0.5, the volume is then padded to its original size. The 0.5 threshold was fixed after a grid-search. Finally, 3D labeling is performed using an implementation from

| Method | Trained on | In-house (DICE %) | ADNI (DICE %) | HARP (DICE %) |
|---|---|---|---|---|
| (Thyreau et al., 2018) | Their data | 86.24 | 74.34 | 85.0 |
| **E2D Consensus** | In-house | **97.22** | **78.66** | 81.39 |
| (Isensee et al., 2017) | In-house | 85.39 | 74.51 | 85.86 |
| **E2D Consensus** | In-house + HARP | 97.00 | 78.18 | 87.36 |
| (Isensee et al., 2017) | HARP | 84.60 | 75.63 | 86.23 |
| **E2D Consensus** | HARP | 90.25 | 77.07 | 87.48 |
| (Ataloglou et al., 2019) | HARP | - | - | **90.15** |

Table 1: Our method is named E2D Consensus. Experimends were made locally, except for (Ataloglou et al., 2019).

(Dougherty and Lotufo, 2003). The two connected labels with more volume are kept, discarding smaller noise volumes.

## 3. Results and Discussion

Our modifications to the U-Net basic architecture improved DICE by around 5% and reduced overfitting when testing in other datasets. Not using batch normalization resulted in poor convergence. Our post processing successfully removes noise from small false positive volumes due to training using patches. The consensus strategy resulted in 2% better performance then evaluation following only one orientation. The best orientation was found to be Coronal or Sagittal depending on the dataset.

Experiments showed state-of-the-art results (Table 1). (Isensee et al., 2017)'s model is a fully 3D UNet approach originally used for brain tumors. (Thyreau et al., 2018)'s pre-registration step failed on HARP. Our model has the best results on ADNI, which was not involved in any model's training. Test data was not included in the models training. Only our data is MNI152 registered, ADNI and HARP are not registered. Our method when trained on public HARP data is superior to some other methods in the literature, FreeSurfer and FSL, as found in the extensive table in (Ataloglou et al., 2019). Results per group of patients for HARP are as follows: CN: 87.66% MCI: 85.28% AD: 84.36%.

Our method has a light memory foot print and used less training volumes than (Thyreau et al., 2018). Segmentation takes around 15 seconds per volume on a mid-range nVidia 1060 GPU. Training of all three networks takes around 12 hours on a Titan X.

## 4. Conclusion

Our main contribution is a novel hippocampus segmentation method that achieves state-of-the-art segmentation performance. Various successful CNN design ideas from the literature were employed in our achitectures, trained in a dataset that is comprised of mostly irregular hippocampus shapes, with results comparable to other recent hippocampus segmentation methods. Our method is available and ready to run in github.com/dscarmo/e2dhipseg.

## Acknowledgments

We thank FAPESP for funding this research under grant 2018/00186-0, our partners at BRAINN (FAPESP number 2013/07559-3 and FAPESP 2015/10369-7) for letting us use their dataset on this research and CNPq research funding, process numbers 310828/2018-0 and 308311/2016-7.

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
