# OpenReview forum: "Extended 2D Consensus Hippocampus Segmentation"
_MIDL.io/2019/Conference/Abstract — MIDL Abstract 2019_

### Official Review · AnonReviewer2 · 2019-04-30
**results need to be extended for a complete comparison**

**Rating:** 2
**Confidence:** 2

**Review:**

The authors proposed an extended 2D segmentation to obtain 3D segmentation for hippocampus data, where consensus segmentation is obtained using 3 extended 2D sagital, coronal and axial segmentations. Proposed method is validated on ADNI and HARP datasets and the authors concluded that it achieved good results compared to fully 3D U-net and other state-of-the-art methods and stated that it also required less memory and time for training.

However the results are not very complete with the following concerns:
1) The method by Ataloglou et al has good performance on HARP but is not available for the other 2 datasets, it will be more complete if the authors can also validate this methods for the other 2 datasets;
2) In table 1, the authors additionally trained the proposed method, but not baseline methods, on private data+HARP, what is the reason for this experiment?

---

### Official Review · AnonReviewer1 · 2019-05-01
**thorough study with good results**

**Rating:** 3
**Confidence:** 2

**Review:**

This abstract employs novel (to my knowledge) approaches for hippocampal segmentation, such as incorporating a consensus across the 3 orientations. Care is taken in pre- and post-processing of the data, and the proposed approach is compared to several alternate approaches and datasets.  Strong results are obtained. As the method performed exceptionally well on the authors’ own data (in comparison to the others), and I was curious how this dataset differed from the other datasets - e.g., with respect to acquisition parameters.

---

### Decision · Program_Chairs · 2019-05-06
**Acceptance Decision**

Accept